# Downsampling leads to Image Memorization in Convolutional Autoencoders

## Abstract

Memorization of data in deep neural networks has become a subject of significant research interest. In this paper, we link memorization of images in deep convolutional autoencoders to downsampling through strided convolution. To analyze this mechanism in a simpler setting, we train linear convolutional autoencoders and show that linear combinations of training data are stored as eigenvectors in the linear operator corresponding to the network when downsampling is used. On the other hand, networks without downsampling do not memorize training data. We provide further evidence that the same effect happens in nonlinear networks. Moreover, downsampling in nonlinear networks causes the model to not only memorize just linear combinations of images, but individual training images. Since convolutional autoencoder components are building blocks of deep convolutional networks, we envision that our findings will shed light on the important phenomenon of memorization in over-parameterized deep networks.

## 1 Introduction and Related Work

As deep convolutional neural networks (CNNs) become ubiquitous in computer vision due to their applicability and strong performance on a range of tasks (Goodfellow et al., 2016), recent work has begun analyzing the memorization properties of such networks in classification. For example, Zhang et al. (2017) show that popular CNNs can achieve almost zero training error on randomly labeled datasets, indicating that CNNs have the capacity to "memorize" large training data sets.

Arpit et al. (2017) and Sablayrolles et al. (2018) build on the experiments from Zhang et al. (2017) to better understand and evaluate the extent to which CNNs memorize training data. Arpit et al. (2017) show that CNNs, when trained on large datasets, are able to learn patterns from realistic data before memorizing training images. Sablayrolles et al. (2018) present experiments on "membership inference" (i.e. determining whether an image was used during training) and conclude that modern architectures are capable of "remember[ing] a large number of images and distinguish[ing] them from unseen images".

Although the above methods analyze memorization in the classification setting, they do not provide a mechanism through which memorization of training data occurs. We here present downsampling as one mechanism by which deep CNNs memorize specific training images. We will focus our study on the memorization properties of linear and nonlinear fully convolutional autoencoders. The architectures we use (such as U-Net, (Ronneberger et al., 2015)) are commonly employed in image-to-image tasks, see e.g. Ulyanov et al. (2017). However, we will use these architectures only in the autoencoding framework. We primarily focus on autoencoders (Baldi, 2012) for the following reasons: (1) components of convolutional autoencoders are building blocks of many CNNs; and (2) layerwise pre-training using autoencoders is a technique to initialize individual layers of CNNs to improve training (Bengio et al. (2007), Erhan et al. (2010)). It is important to note that there are many potential solutions to the autoencoding problem when using over-parameterized autoencoders. In particular, in the linear case, these models may range from learning the (full rank) identity function (which has 0 error in the autoencoding task) to low rank solutions where each training example corresponds to an eigenvector with eigenvalue 1. Thus, understanding how autoencoders learn is of interest in order to gain insights into how deep CNNs memorize training data.

Figures 1a and 1b provide two examples of memorization: A typical U-Net architecture (the same as e.g. used in Ulyanov et al. (2017) for large hole impainting) when trained on a single image

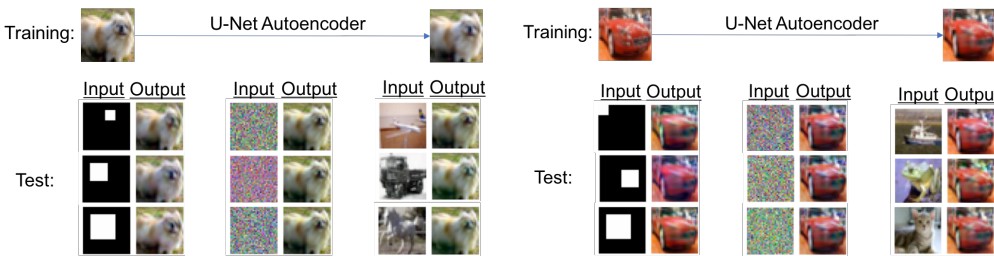

(a) U-Net Autoencoder trained on a dog.    (b) U-Net Autoencoder trained on a car.

Figure 1: U-Net Autoencoder trained on a single image from CIFAR10 for 2000 iterations. When fed random sized white squares, a standard Gaussian, or new images from CIFAR10 the model outputs the training image.

"memorizes" the training image in the sense that for any input, the output always contains the training image (even if the input is random noise or an arbitrary white square). This paper provides a mechanism for this phenomenon.

The outline is as follows: After introducing some notation in Section 2, we will show in Section 3 that memorization is tightly coupled with downsampling and also occurs in the simpler setting of *linear* autoencoding CNNs. In the linear setting, the neural network corresponds to matrix multiplication. In Section 4, we show how to extract this matrix representation and we provide our main conjecture, namely that linear combinations of the training images are stored as eigenvectors of this matrix, whose rank is given by the dimension of the span of the training set. We also provide strong evidence for this conjecture on $2 \times 2$ images. In Section 5, we analyze the eigenvalue decay and show in various examples that using downsampling linear CNNs, linear combinations of the training examples are stored as eigenvectors with eigenvalues close to $1$. Finally, we return to the nonlinear setting in Section 6, providing evidence that memorization is an even stronger phenomenon in nonlinear networks, since the actual training images (in contrast to linear combinations of training images) are memorized. We end with a short discussion in Section 7.

## 2 NOTATION

In this section, we introduce the mathematical framework for our work and highlight two different functions learned by autoencoding CNNs, namely the identity function and the point map.

We denote a training set of $n$ square images by $X = \{x_1, x_2, ..., x_n\}$, where each $x_i \in \mathbb{R}^{c \times s \times s}$ with $c$ being the number of color channels (or filter channels) and $s$ denoting the width and height of an image. We will focus on *convolutional autoencoders*, i.e., CNNs trained to map between the same image. In particular, we will consider *linear CNNs*, by which we mean convolutional autoencoders with layers being either nearest neighbor upsampling or convolutional with kernel size 3, zero padding, no activation functions, and no biases. To simplify the computations in Section 4, we assume throughout that the input image as well as the stride size are a power of 2.

We denote the function learned by a CNN on the training set $X$ by $\mathcal{C}_X : \mathbb{R}^{c \times s \times s} \to \mathbb{R}^{c \times s \times s}$. The training procedure minimizes a given loss function between the input image and its reconstruction by $\mathcal{C}_X$. We use the mean squared error loss throughout; thus the loss function is given by $\sum_{i=1}^{i=n} \frac{1}{ncs^2} (\mathcal{C}_X(x_i) - x_i)^2$, where subtraction and exponentiation are taken elementwise. For linear CNNs, we denote the matrix corresponding to $\mathcal{C}_X$ by $A_X$. Denoting the vectorized (or "flattened") version of an image $y \in \mathbb{R}^{c \times s \times s}$ by $y_f \in \mathbb{R}^{cs^2}$, then $A_X$ satisfies $A_X y_f = (\mathcal{C}_X(y))_f$.

In this work, we identify and analyze an architectural mechanism that is able to fundamentally alter the function $\mathcal{C}_X$ learned by an autoencoding CNN on a given training set. The following two functions will play an important role in the subsequent analysis: (1) the *identity function* given by $\mathcal{C}_X(y) = y$ for any $y \in \mathbb{R}^{c \times s \times s}$ and (2) the *point map* given by $\mathcal{C}_X(y) = \mathcal{C}_X(x_0)$ for any $y \in \mathbb{R}^{c \times s \times s}$, where $x_0$ is a particular element in $\operatorname{span} \mathcal{C}_X(X)$ where $\operatorname{span} \mathcal{C}_X(X) = \operatorname{span} \{\mathcal{C}_X(x_i), 1 \le i \le n\}$. An extreme form of *memorization* occurs when a CNN learns the point map on a training set of size one, i.e., it maps *any* image to the *same fixed* image.

## 3 DOWNSAMPLING: A MECHANISM FOR MEMORIZATION

In this section, we will illustrate how downsampling acts as a mechanism for learning the point map in deep CNNs. In particular, we will show that even if a downsampling network has the capacity to learn the identity function, it prefers to learn the point map.

We consider the linear CNNs defined by Network ND (non-downsampling) and Network D (downsampling) in Figure 2a. Both networks employ 128 filters in each convolutional layer in all but the last layer, which contains 3 filters. The two networks only differ by the property that Network D uses filters of stride 8 in the first layer to *downsample* the image and then later uses nearest neighbor upsampling with scale factor 8 to rescale the image, while Network ND does not perform any downsampling. In concordance with the severe downsampling from $224 \times 224$ to $7 \times 7$ images performed by ResNet and VGG (He et al. (2016); Simonyan & Zisserman (2015)), Network D downsamples from $32 \times 32$ to $4 \times 4$ images. However, we will show in Section 5 that the same memorization phenomenon is also observed in networks that only downsample using multiple convolutional layers of stride 2 as is done in U-Nets (Ronneberger et al. (2015)). In the examples, the networks were initialized using the default in the PyTorch deep learning library (Paszke et al., 2017), namely each layer's weights were drawn i.i.d. from a uniform distribution $\mathcal{U}(-a, a)$ with $a = 1/\sqrt{P}$ where $P$ is the number of parameters in that layer.

Both networks were trained for 2000 iteration on a single $3 \times 32 \times 32$ image from CIFAR10 (Krizhevsky, 2009) of a frog. Network ND reconstructed the digit 3 with a training loss of $10^{-4}$ and Network ND with loss $10^{-2}$. We then applied the trained networks to test images consisting of both realistic images and random images with each pixel in the image being drawn from the standard normal distribution. As shown in Figure 2c and 2b, although trained on the same input for the same number of iterations Network D (with downsamling) learned the point map while Network ND (no downsampling) learned the identity function up to sign. The possible sign flip is a consequence of using no biases. In fact, this contrasting behavior is very robust throughout the training process. In Figure 3, we see that regardless of when we stop training, even after only 10 iterations, Network D learns a point map, while Network ND learns a function visually similar to the identity map.

One may conjecture that the memorization phenomenon that occurs using Network D could be due to the small number of filters in the downsampling layer, so that not all the features from the input could be preserved and the network would not be capable of learning a full rank identity map. This is not the case. To demonstrate that this is in general not the reason why downsampling networks learn a point map, we next present a downsampling network which, as we show, has the capacity to learn the identity function, but still prefers to learn the point map instead as the result of optimization.

**Example.** *Consider the linear CNN defined by the small downsampling Network DS presented in Figure 4a, which is trained on a single $1 \times 2 \times 2$ image for $2000$ iterations. Figure 4b shows that there exists an initialization such that the network learns the identity function. A full description of how we computed this manual initialization is provided in Appendix A. In contrast, as shown in Figure 4c initializing Network DS with the default PyTorch uniform distribution results in the point map.*

This provides the first example showing that even though the network has the capacity to learn the identity function, it prefers to learn a point map using the default PyTorch initialization. We observed the same effects using the Xavier uniform initialization, which is based on the assumption

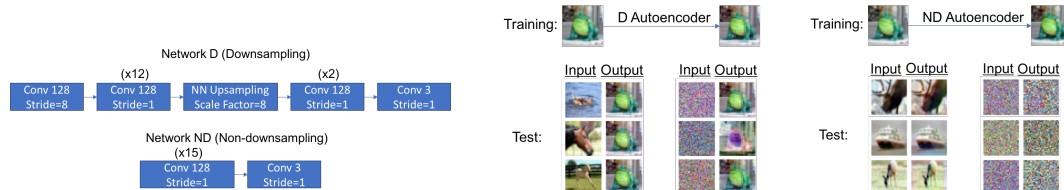

(a) Downsampling (D) and non-downsampling (ND) networks

(b) Example with Network D  (c) Example with Network ND

Figure 2: Linear CNNs with Network ND (non-downsampling) and Network D (downsampling) were trained for 2000 iterations to autoencode a single frog from CIFAR10. While Network ND approximately learns the identity function, Network D learns the point map (up to sign).

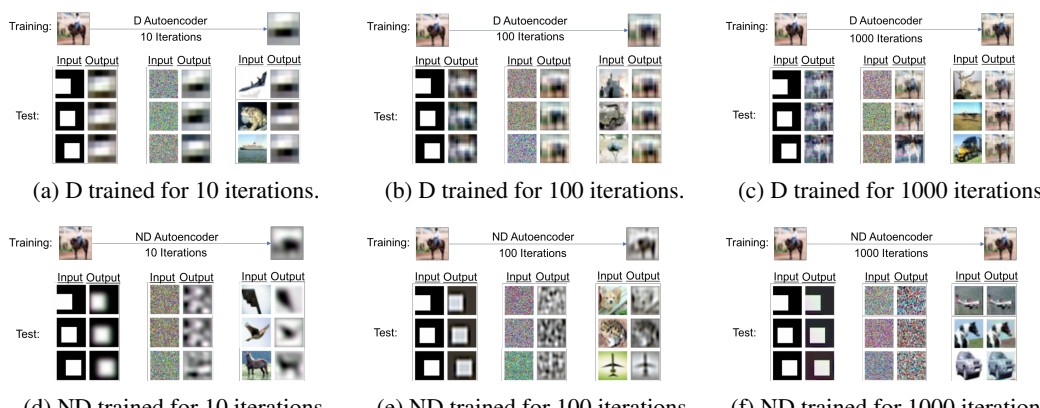

(a) D trained for 10 iterations.  (b) D trained for 100 iterations.  (c) D trained for 1000 iterations.

(d) ND trained for 10 iterations.  (e) ND trained for 100 iterations.  (f) ND trained for 1000 iterations.

Figure 3: While the downsampling network D learns the point map regardless of the stopping point, the non-downsampling network ND begins to learn a function similar to the identity even after training on a single image for only 10 iterations.

that activations are linear (Glorot & Bengio (2010)). However, the results are not observed for linear networks when using Kaiming initialization, which is expected since this initialization is meant for nonlinear networks with ReLU/PReLU activations (He et al. (2015)). In Appendix E, we show that nonlinear networks memorize training images under any of these initializations.

# 4 MATHEMATICAL ANALYSIS OF MEMORIZATION IN LINEAR CNNS

We now turn to extracting and analyzing the linear operator that describes the function learned by the linear CNN. While the full algorithms for extracting the matrix $A_X$ from the network are described in Appendix B, we here provide intuition for this procedure. Since the matrix $A_X$ can be decomposed as a product of matrices, one for each layer, it suffices to provide intuition for converting a general convolutional layer and a general upsampling layer into a matrix. We then discuss our main conjecture and proposition linking the eigenvectors of $A_X$ to the mechanism of memorization.

## 4.1 EXTRACTING THE LINEAR OPERATOR FROM A LINEAR CNN

To simplify notation, we assume that the inputs and outputs are already zero padded images (with 1 layer of zero padding on each side), i.e., the original images lie in $\mathbb{R}^{c \times (s-2) \times (s-2)}$ such that the images after vectorization and zero padding lie in $\mathbb{R}^{cs^2}$. The convolutions all consist of kernels of size $3 \times 3$, 1 layer of zero padding. Consider a particular convolutional layer consisting of $f$ filters,

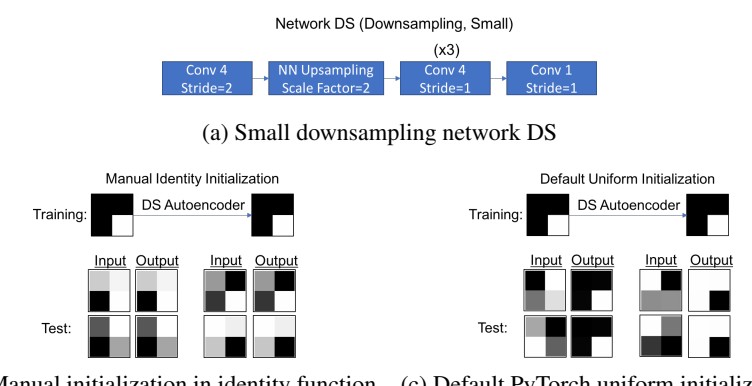

(a) Small downsampling network DS

(b) Manual initialization in identity function    (c) Default PyTorch uniform initialization

Figure 4: Linear CNN with network DS when trained on a single $2 \times 2$ image learns the identity function when manually initialized in the identity function, but learns the point map when initiated using the default PyTorch uniform initialization.

stride size $t$, kernel volume $f' \times 3 \times 3$, and 1 layer of zero padding, which operates on an incoming zero padded vectorized image of $f's'^2$ voxels, where $f'$ is the depth and $s'$ the width and height of the incoming image. Assuming that the stride and image sizes are powers of 2, it follows that $(s'-2)$ is a multiple of $t$. Hence the matrix corresponding to the operator of this particular layer is of size $f(\frac{s'-2}{t}+2)^2 \times f's'^2$, where $\frac{s'-2}{t}+2$ corresponds to the width and height of the output after striding and zero padding. The matrix itself is circulant-like, where the zero padding must be carefully accounted for by additional shifts and rows of zeros. A particular example is shown in Appendix B.

Next, we provide intuition on how to extract the matrix corresponding to the nearest neighbor up-sampling operation. The full algorithm together with an example is given in Appendix B. Given a vectorized zero-padded image of $f's'^2$ voxels as input, then nearest neighbor upsampling with scale factor $k$ corresponds to a matrix of size $f'(k(s'-2)+2)^2 \times f's'^2$, where $(k(s'-2)+2)$ comes from having to scale only the non-zero-padded elements of the representation and adding 2 for the new zero padding. This matrix is composed of blocks of circulant-like matrices consisting of one-hot vectors, where blocks of $s$ identical rows are shifted by 1 element. Finally, the full operator $A_X$ for a linear network $\mathcal{C}_X$ is obtained by multiplying the matrices corresponding to each layer.

## 4.2 Spectral Analysis of Linear Operator

Being able to extract the linear operator $A_X$ from a linear CNN is critical for analyzing the mechanism of memorization, since it allows an examination of its eigenvalues and eigenvectors. Note that $A_X$ operates on the space of $c \times s \times s$ images. Hence every eigenvector (when stripped of zero-padding and reshaped to $c \times s \times s$) represents an image in this space. However, since $A_X$ is in general not symmetric, eigenvectors can be complex and hence the real and imaginary components of the eigenvectors represent separate images in the original space.

We can easily obtain a bound on the rank of $A_X$: Since $A_X$ is given by a product of matrices, one for each layer in the linear CNN, the rank of $A_X$ is at most equal to the rank of the minimal rank matrix in the factorization of $A_X$. We denote this number by $r_{min}$. We now present our main conjecture.

**Conjecture.** *Let* $X = \{x_1, \ldots, x_n\}$, $x_i \in \mathbb{R}^{c \times s \times s}$, *be a training set of images of size* $c \times s \times s$.

(a) *If* $A_X$ *is the linear operator of a* downsampling *network trained to zero loss, then*

$$rank(A_X) = \min\left(dim(\text{span}\,(X), r_{min})\right),$$

*where all eigenvalues are equal to 1 and the corresponding eigenvectors are linear combinations of the input training images.*

(b) *If* $A_X$ *is the linear operator of a* non-downsampling *network trained to zero loss, then*

$$\min\left(dim(\text{span}\,(X), r_{min})\right) \leq rank(A_X) \leq cs^2$$

In other words, we conjecture that linear downsampling networks learn a low rank solution when the number of linearly independent training examples is not sufficiently large (i.e. larger than $cs^2$), even when they have the capacity to learn the identity function. In particular, we conjecture that the rank is given by the dimension of the span of the training images. Most importantly, we conjecture that the mechanism of memorization in downsampling networks is by storing linear combinations of the training images as eigenvectors of the linear operator. On the other hand, we conjecture that linear non-downsampling networks learn a much higher rank solution, thereby often rendering the solution visually indistinguishable from the identity function. Our conjecture also implies that when training a linear downsampling CNN on images of size $3 \cdot 224 \cdot 224$, which corresponds to the input image size for VGG and ResNet (He et al. (2016), Simonyan & Zisserman (2015)), the number of linearly independent training examples needs to be at least $3 \cdot 224 \cdot 224 = 153,228$ before the network can learn the identity function. However, assuming that realistic images lie on a low-dimensional manifold, this conjecture indicates that a linear downsampling network will learn a basis for the low-dimensional manifold instead of the identity when trained on sufficiently many realistic images.

Before providing empirical evidence for our conjecture in Section 5, we end this section by summarizing our theoretical evidence in the following proposition showing that the network in Figure 4a can learn solutions of all ranks 1 through 4 for $1 \times 2 \times 2$ images depending on the dimension of the span of the training set.

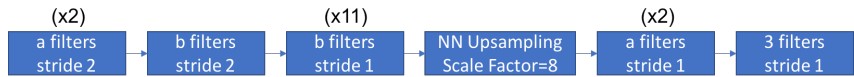

Figure 5: The general architecture of a linear CNN with $a$ filters on upsampled representations and $b$ filters on downsampled representations. We denote the downsampling version of this network as Network $D_{a,b}$ and the non-downsampling version Network $ND_{a,b}$.

**Proposition.** *The linear Network DS presented in Figure 4a can learn a linear operator of rank* $\min\left(dim(\text{span}\,(X), 4)\right)$ *with* $1 \leq dim(\text{span}\,(X)) \leq 4$.

The proof is given in Appendix C. Note that this proposition does not imply that $A_X$ *must* obtain rank $r = \min\left(\dim\left(\text{span}\,(X), 4\right)\right)$ for all training sets $X$ on $1 \times 2 \times 2$ images, but rather that the conjecture holds for any example of training sets that we tried, in particular also training sets covering each possible nonzero value of $\min\left(\dim\left(\text{span}\,(X), 4\right)\right)$.

## 5 EMPIRICAL EVIDENCE FOR MAIN CONJECTURE

We now provide empirical evidence for our main conjecture by analyzing the operator $A_X$ for networks trained on CIFAR10 color images (which have width and height 32). First, we show that when training on one or two images respectively, a downsampling network learns an operator that has very low effective rank with the top eigenvector(s) corresponding to the training image(s). On the other hand, we show that a non-downsampling network trained on the same data learns an operator with much higher rank, closer to the full dimensionality of the data. Finally, we will train downsampling networks on tens to thousands of images from a single class of CIFAR10 and present the rank of the learned solutions in Appendix D.

All of our downsampling networks downsample using 3 convolutional layers with stride 2 and then upsample using a 1 nearest neighbor upsampling layer with scale factor 8. The general architecture is shown in Figure 5. Importantly, we note that if we use such a network on color CIFAR10 images, we need at least $b = 3 \cdot 8 \cdot 8 = 192$ filters in order to be able to learn the identity function since otherwise $r_{min} < 3 \cdot 32 \cdot 32$. Our non-downsampling networks have the same convolutional layer scheme as the downsampling networks. We denote a downsampling (respectively non-downsampling) network with $a$ filters on upsampled representations and $b$ filters on downsampled representations by $D_{a,b}$ (respectively $ND_{a,b}$).

The networks are initialized using the default initialization in PyTorch (Paszke et al. (2017)), trained using the Adam optimizer (Kingma & Ba (2015)) with learning rate $10^{-4}$, and trained until the loss decreased by less than $10^{-4}$ or for $20,000$ epochs. Note that we disregard overfitting in training, since we expect an overfitted autoencoder to learn the identity function. As pointed out in Figure 3, the memorization effect is robust throughout the training process. However, in this case, we wish to train until the training error is sufficiently low (i.e. around $10^{-3}$) so that we can understand the rank of the operator in relation to our conjecture.

Figure 6 shows the results for various downsampling and non-downsampling networks when trained on $1 \leq k \leq 2$ images. As shown in Figures 6a and 6b, in both cases Network $D_{64,192}$ stores the $k$ images (up to a shift in the color channels) as eigenvectors with eigenvalues close to 1 and produces roughly a rank $k$ solution. In particular, when trained on a single picture of a plane from CIFAR10 this network stores the training image as the real component of the top eigenvector; it is an inverted color image of the plane and the rank of the learned solution is approximately 1 (Figure 6a). When trained on a picture of the plane and a car, the top two eigenvectors are linear combinations of the plane and the car with the plane having inverted colors (Figure 6b). In fact, the plane is more visible in the complex component of the second eigenvector while the car is more visible in the real components. The learned solution is of rank approximately 2 with the top two eigenvalues being complex conjugates (hence having the same magnitude). On the other hand, Figures 6c and 6d show that training the non-downsampling network $ND_{4,4}$ on a single image or on two images results in a higher-rank solution, with rank and spectrum being independent of $k$.

When training on $k > 2$ images from different classes of CIFAR10, our conjecture stipulates that the rank of the learned solution indicates the dimension of the span of the training set. Since visu-

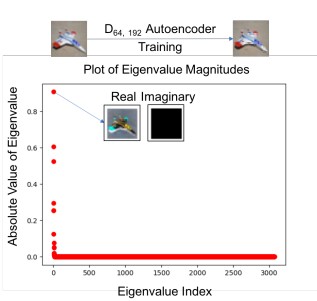

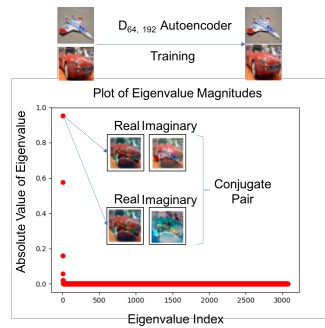

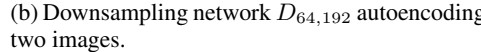

(a) Downsampling network $D_{64,192}$ autoencoding a single image.

(b) Downsampling network $D_{64,192}$ autoencoding two images.

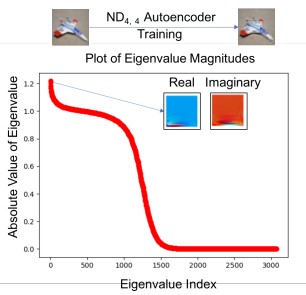

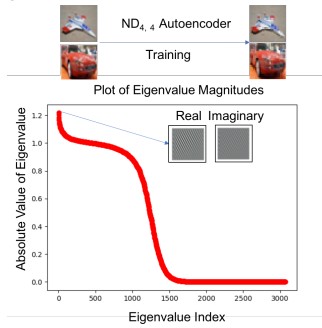

(c) Non-downsampling network $ND_{4,4}$ autoencoding a single image.

(d) Non-downsampling network $ND_{4,4}$ autoencoding two images.

Figure 6: Plots show magnitude of eigenvalues together with visualization of real and imaginary components of the top eigenvector and their complex conjugates (for complex eigenvalues).

alization of the eigenvectors when trained on $k > 2$ images is not as insightful (it becomes difficult to visually identify the training images in the linear combination), the corresponding visualizations and eigenvalue plots are moved to Appendix D.

## 6 MEMORIZATION IN NON-LINEAR AUTOENCODERS

In this section, we analyze the phenomenon of memorization for non-linear networks. In the following experiments, we use the linear CNNs $D_{a,b}$ and $ND_{a,b}$ as the backbone architecture but with LeakyReLU activations (Xu et al., 2015) after every convolutional layer. We denote the resulting architectures by $NLD_{a,b}$ (for the non-linear downsampling network with backbone $D_{a,b}$) and $NLND_{a,b}$ (for the non-linear non-downsampling network with backbone $ND_{a,b}$).

Figure 7 shows the impact of downsampling on nonlinear networks trained on a single image. It is apparent that $NLD_{128,128}$ has memorized the training image while $NLND_{128,128}$ has not, since for artifical, random noise, and real image inputs $NLD_{128,128}$ always outputs the training example while $NLND_{128,128}$'s output is more simliar to the identity function.

For Figure 8 we trained the non-linear downsampling network $NLD_{128,128}$ on a set of 10 images, one from each class of CIFAR10. Interestingly, when fed in a new image from CIFAR10, Gaussian noise or artificial white squares, the output was always one of the training images. This suggests that the effect of memorization is even stronger for non-linear downsampling networks, since they output specific training examples as compared to linear combinations of training examples, which we had observed for linear downsampling networks. In other words, the individual training examples act as strongly attracting fixed points for non-linear downsampling networks.

We end this section with a remark on the impact of *different initializations* on training. While the experiments shown in Figure 7 were initialized in the default uniform distribution from PyTorch, Appendix E shows results with different initializations. In particular, we show that memorization still occurs when training $NLD_{a,b}$ with either Xavier or Kaiming uniform/normal initializations.

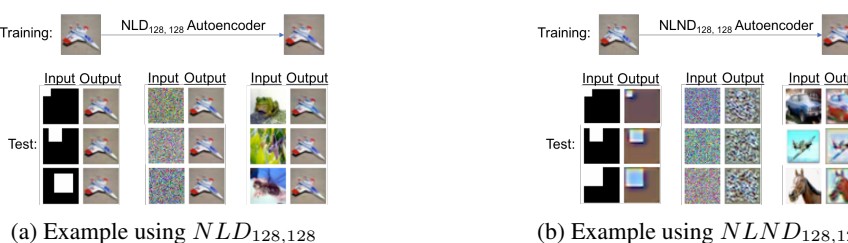

(a) Example using $NLD_{128,128}$          (b) Example using $NLND_{128,128}$

Figure 7: A comparison of nonlinear non-dowsampling ($NLND_{128,128}$) and nonlinear down-sampling ($NLD_{128,128}$)) networks when trained on a single image of a plane from CIFAR10. $NLD_{128,128}$ has learned the point map while $NLND_{128,128}$ has not.

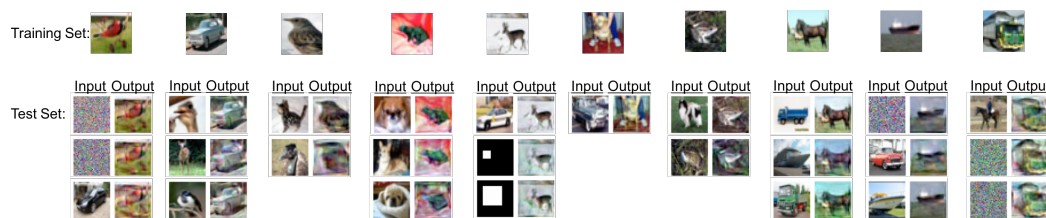

Figure 8: The nonlinear downsampling network $NLD_{128,128}$ is trained on a set of 10 images, 1 from each class of CIFAR10. The training images are reproduced when feeding in new test images from CIFAR10, white squares or standard normal noise. For each of the training examples, we present some test examples that produced the training example. Interestingly, only the deer was reproduced when given artificial images of white squares.

## 7    CONCLUSIONS AND FUTURE WORK

This paper identified downsampling as a mechanism through which linear CNNs memorize training images. We demonstrated that downsampling convolutional autoencoders memorize training images in both the linear and nonlinear setting. In particular, we showed that it is not just the dimensionality reduction of downsampling that causes these models to learn point maps by demonstrating that a downsampling CNN architecture with the capacity to learn the identity function still prefers the point map. In the linear case, this preference for low-rank over the equally valid high-rank solutions is highly suggestive of similar phenomena observed in problems such as matrix completion (e.g.,Gunasekar et al.).

In the non-linear case, memorization in downsampling networks is manifested even more strikingly with nearly arbitrary input images being mapped to output images that are visually identifiable as one of the training images. While the exact mechanism still needs to be explored, this is reminiscent of FastICA in Independent Component Analysis (Hyvrinen & Oja, 1997) or more general non-linear eigen-problems (Belkin et al., 2018), where every "eigenvector" for certain iterative maps has its own basin of attraction. On the other hand, non-downsampling auto-encoders do not memorize the training data and consistently learn a "high rank" map, similar to the identity map, at least visually.

We conjecture that our findings will help to shed light on the strong generalization properties of downsampling networks for image classification and recognition tasks. Indeed, if downsampling networks memorize images or linear combinations of images, when trained on large datasets, they may be capable of learning representations within the space of all realisitic images instead of learning the standard full rank basis.

We conclude with a mention of further areas of exploration spurred on by our work. We still need to understand why downsampling forces the network to learn low rank solutions even when the network has the capacity to learn the identity. This requires developing a better grasp of optimization and initialization, starting with linear autoencoders and proceeding to the non-linear settings. Finally, we need to explore connections between our conjecture and the manifold hypothesis to better understand the space of realistic images.

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

# APPENDIX

## A  Manual Initialization for Small Networks

Here we provide the initialization of the identity function for the network from Figure 4a. The procedure is as follows (the positions are 0 indexed):

1. There are 4 filters in the first layer each of size $(3, 3)$ and with stride 2. Filter 1 should have indices $(1, 1)$ set to 1. Filter 2 should have indices $(1, 2)$ set to 1. Filter 3 should have indices $(2, 1)$ set to 1. Filter 4 should have indices $(2, 2)$ set to 1.

2. Now nearest neighbor upsampling will be applied to each of the outputs.

3. The second convolutional layer has 4 filters of size $(4, 3, 3)$. Filter $i$ should have position $(i, 1, 2)$ set to 1 for $0 \le i \le 3$.

4. The third convolutional layer has 4 filters of size $(4, 3, 3)$. Filter $i$ should have position $(i, 2, 1)$ set to 1 for $0 \le i \le 3$.

5. The fourth convolutional layer has 4 filters of size $(4, 3, 3)$. Filter 1 should have indices $(0, 1, 1)$ set to 1. Filter 2 should have indices $(1, 1, 0)$ set to 1. Filter 3 should have indices $(2, 0, 1)$ set to 1. Filter 4 should have indices $(3, 0, 0)$ set to 1.

6. The last filter is of size $(4, 3, 3)$. This filter should have indices $(i, 1, 1)$ set to 1 for $0 \le i \le 3$.

## B  Linearizing a CNN

In this section, we present algorithms for converting convolutional layers and nearest neighbor upsampling layers into matrices. We first present how to construct a block of this matrix for a single filter in Algorithm 1. To construct a matrix for multiple filters, one need only apply the provided algorithm to construct separate matrix blocks for each filter and then concatenate them. We now provide an example of how to convert the first layer from the network in Figure 4a into a single matrix for $1 \times 2 \times 2$ images.

First suppose we have a zero padded $1 \times 2 \times 2$ matrix as input, which is shown vectorized to the right:

$$\begin{bmatrix} 0 & 0 & 0 & 0 \\ 0 & A_1 & A_2 & 0 \\ 0 & A_3 & A_4 & 0 \\ 0 & 0 & 0 & 0 \end{bmatrix} \rightarrow \begin{bmatrix} 0 & 0 & 0 & 0 & 0 & A_1 & A_2 & 0 & 0 & A_3 & A_4 & 0 & 0 & 0 & 0 & 0 \end{bmatrix}^T$$

Now we apply the first convolutional layer to this $16 \times 1$ vector. We have 4 convolutional filters with parameter $C_i^{(j)}$ denote the $i$th parameter (read in row-major order) from filter $j$ for $1 \le i \le 9$ and $1 \le j \le 4$. The resulting convolutional matrix is of size $4 \cdot 3 \cdot 3 \times 1 \cdot 4 \cdot 4$, but here we show just the first 9 columns (i.e. the pattern for the $i$th filter):

$$\begin{bmatrix} 0 & 0 & 0 & 0 & 0 & 0 & 0 & 0 & 0 & 0 & 0 & 0 & 0 & 0 & 0 & 0 \\ 0 & 0 & 0 & 0 & 0 & 0 & 0 & 0 & 0 & 0 & 0 & 0 & 0 & 0 & 0 & 0 \\ 0 & 0 & 0 & 0 & 0 & 0 & 0 & 0 & 0 & 0 & 0 & 0 & 0 & 0 & 0 & 0 \\ 0 & 0 & 0 & 0 & 0 & 0 & 0 & 0 & 0 & 0 & 0 & 0 & 0 & 0 & 0 & 0 \\ 0 & 0 & 0 & 0 & 0 & C_5^{(i)} & C_6^{(i)} & 0 & 0 & C_8^{(i)} & C_9^{(i)} & 0 & 0 & 0 & 0 & 0 \\ 0 & 0 & 0 & 0 & 0 & 0 & 0 & 0 & 0 & 0 & 0 & 0 & 0 & 0 & 0 & 0 \\ 0 & 0 & 0 & 0 & 0 & 0 & 0 & 0 & 0 & 0 & 0 & 0 & 0 & 0 & 0 & 0 \\ 0 & 0 & 0 & 0 & 0 & 0 & 0 & 0 & 0 & 0 & 0 & 0 & 0 & 0 & 0 & 0 \\ 0 & 0 & 0 & 0 & 0 & 0 & 0 & 0 & 0 & 0 & 0 & 0 & 0 & 0 & 0 & 0 \end{bmatrix}$$

We similarly create matrices for the other convolutional layers.

Next we present how to construct the rows of such a matrix in Algorithm 2.

---

**Algorithm 1** Create Matrix for Single Convolutional Filter given Input with dimensions $f \times s \times s$

---

**Input:** $parameters$:= parameters of $f$ trained $3 \times 3$ CNN filters, $s$:= width and height of image without zero padding, $f$:= depth of image, $stride$:= stride of CNN filter
**Output:** Matrix $C$ representing convolution operation

1: **function** CREATEFILTERMATRIX($parameters, s, f, stride$)
2:      $paddedSize \leftarrow s + 2$
3:      $resized \leftarrow s/stride$
4:      $rowBlocks \leftarrow$ zeros matrix size $(f, (paddedSize)^2)$
5:      **for** $filterIndex \leftarrow 0$ to $f - 1$ **do**
6:         **for** $kernelIndex \leftarrow 0$ to $8$ **do**
7:             $rowIndex \leftarrow kernelIndex \mod 3 + paddedSize \lfloor \frac{kernelIndex}{3} \rfloor$
8:             $rowBlocks[filterIndex][rowIndex] \leftarrow parameters[filterIndex][kernelIndex]$
9:         **end for**
10:      **end for**
11:      $C \leftarrow$ zeros matrix of size $((resized + 2)^2, f \cdot paddedSize^2)$
12:      $index \leftarrow resized + 2 + 1$
13:      **for** $shift \leftarrow 0$ to $resized - 1$ **do**
14:         $nextBlock \leftarrow$ zeros matrix of size $(resized, f \cdot paddedSize^2)$
15:         $nextBlock[0] \leftarrow rowBlocks$
16:         **for** $rowShift \leftarrow 1$ to $resized - 1$ **do**
17:             $nextBlock[rowShift] \leftarrow rowBlocks$ shifted right by $stride \cdot rowShift$
18:         **end for**
19:         $C[index : index + resized, :] \leftarrow nextBlock$
20:         $index \leftarrow index + resize + 2$
21:         $rowBlock \leftarrow$ zero shift $rowBlock$ by $paddedSize \cdot stride$
22:      **end for**
23:      **return** $C$
24: **end function**

---

We now provide the upsampling matrix for an upsampling layer with scale factor 2 operating on a vectorized zero padded $1 \times 1 \times 1$ image:

$$\begin{bmatrix} 0 & 0 & 0 & 0 & 0 & 0 & 0 & 0 & 0 \\ 0 & 0 & 0 & 0 & 0 & 0 & 0 & 0 & 0 \\ 0 & 0 & 0 & 0 & 0 & 0 & 0 & 0 & 0 \\ 0 & 0 & 0 & 0 & 0 & 0 & 0 & 0 & 0 \\ 0 & 0 & 0 & 0 & 0 & 0 & 0 & 0 & 0 \\ 0 & 0 & 0 & 0 & 1 & 0 & 0 & 0 & 0 \\ 0 & 0 & 0 & 0 & 1 & 0 & 0 & 0 & 0 \\ 0 & 0 & 0 & 0 & 0 & 0 & 0 & 0 & 0 \\ 0 & 0 & 0 & 0 & 0 & 0 & 0 & 0 & 0 \\ 0 & 0 & 0 & 0 & 1 & 0 & 0 & 0 & 0 \\ 0 & 0 & 0 & 0 & 1 & 0 & 0 & 0 & 0 \\ 0 & 0 & 0 & 0 & 0 & 0 & 0 & 0 & 0 \\ 0 & 0 & 0 & 0 & 0 & 0 & 0 & 0 & 0 \\ 0 & 0 & 0 & 0 & 0 & 0 & 0 & 0 & 0 \\ 0 & 0 & 0 & 0 & 0 & 0 & 0 & 0 & 0 \\ 0 & 0 & 0 & 0 & 0 & 0 & 0 & 0 & 0 \end{bmatrix}$$

We now give an example for the network from Figure 3a when it is applied to vectorized zero padded $1 \times 2 \times 2$ input images. Suppose that the $i$th convolutional layer can be written as a matrix $A_i$ and the upsampling layer is written as a matrix $U$. Then we have the following matrix factorization for our network:

$$A_X = A_5 A_4 A_3 A_2 U A_1$$

Now $A_1$ has dimensions $4 \cdot 3 \cdot 3 \times 1 \cdot 4 \cdot 4$. $U$ has dimensions $4 \cdot 4 \cdot 4 \times 4 \cdot 3 \cdot 3$. Now $A_2, A_3, A_4$ have dimensions $4 \cdot 4 \cdot 4 \times 4 \cdot 4 \cdot 4$. Finally $A_5$ has dimensions $1 \cdot 4 \cdot 4 \times 4 \cdot 4 \cdot 4$. Hence $A_X$ has

---

**Algorithm 2** Create Matrix for Nearest Neighbor Upsampling Layer

---

**Input:** $s$:= width and height of image without zero padding, $f$:= depth of image, $scale$:= re-scaling factor for incoming image
**Output:** Matrix $U$ representing convolution operation

1: **function** CREATEUPSAMPLINGMATRIX($s, f, scale$)
2:     $outputSize \leftarrow s \cdot scale + 2$
3:     $U \leftarrow$ zeros matrix of size $(f \cdot outputSize^2, f \cdot (s+2)^2)$
4:     $index \leftarrow outputSize + 1$
5:     **for** $filterIndex \leftarrow 0$ to $f - 1$ **do**
6:         **for** $rowIndex \leftarrow 1$ to $s$ **do**
7:             **for** $scaleIndex \leftarrow 0$ to $scale - 1$ **do**
8:                 **for** $columnIndex \leftarrow 0$ to $s$ **do**
9:                     $row \leftarrow$ zeros vector of size $(f(s+2)^2)$
10:                    $row[columnIndex + rowIndex(s+2) + filterIndex(s+2)^2] \leftarrow 1$
11:                    **for** $repeatIndex \leftarrow 0$ to $scale - 1$ **do**
12:                       $U[index] \leftarrow row$
13:                       $index \leftarrow index + 1$
14:                    **end for**
15:                 **end for**
16:                 $indexindex + 2$
17:             **end for**
18:         **end for**
19:         $index \leftarrow index + 2 \cdot outputSize$
20:     **end for**
21:     **return** $U$
22: **end function**

---

dimensions $1 \cdot 4 \cdot 4 \times 1 \cdot 4 \cdot 4$. Now the rank of $A_X$ is at most the rank of any of $A_1, A_2, A_3, A_4, A_5, U$. Furthermore, as $A_X$ is operating on vectorized zero-padded images, the rank of $A_X$ is at most the size of the vectorized original image without zero padding. For this example, this means that the rank of $A_X$ is at most $4 = 1 \cdot 2 \cdot 2$.

## C    PROOF OF PROPOSITION

Here we present the proof of our proposition. We first show that $A_X$ obtains rank $k = \dim(\text{span}(X))$ when $X$ is a nonzero subset of $k$ of the following basis elements for the space $1 \times 2 \times 2$:

$$\begin{bmatrix} 1 & 0 \\ 0 & 0 \end{bmatrix}, \begin{bmatrix} 0 & 1 \\ 0 & 0 \end{bmatrix}, \begin{bmatrix} 0 & 0 \\ 1 & 0 \end{bmatrix}, \begin{bmatrix} 0 & 0 \\ 0 & 1 \end{bmatrix}$$

1. We first train Network DS on

$$\begin{bmatrix} 1 & 0 \\ 0 & 0 \end{bmatrix}$$

After training, the resulting eigenvalues are $1, -7.2 \cdot 10^{-4} + 0.003i, -7.22 \cdot 10^{-4} - 0.003i, -1 \cdot 10^{-2}$. The corresponding largest eigenvector is

$$\begin{bmatrix} -1 & 0 \\ 0 & 0 \end{bmatrix}$$

which is the negative of the training example.

2. We now train Network DS on

$$\begin{bmatrix} 1 & 0 \\ 0 & 0 \end{bmatrix}, \begin{bmatrix} 0 & 1 \\ 0 & 0 \end{bmatrix}$$

After training, the resulting eigenvalues are $1, 1, 9.1 \cdot 10^{-4}, -6.3 \cdot 10^{-3}$. Now the top 2 eigenvectors are linear combinations of the training examples:

$$\begin{bmatrix} .21 & .98 \\ 0 & 0 \end{bmatrix}, \begin{bmatrix} -.67 & .7 \\ 0 & 0 \end{bmatrix}$$

3. We now train DS on

$$\begin{bmatrix} 1 & 0 \\ 0 & 0 \end{bmatrix}, \begin{bmatrix} 0 & 1 \\ 0 & 0 \end{bmatrix}, \begin{bmatrix} 0 & 0 \\ 1 & 0 \end{bmatrix}$$

After training, the resulting eigenvalues are $1, 1, 1, 0.001$. The top 3 eigenvectors are:

$$\begin{bmatrix} .78 & -.43 \\ 0.003 & 0 \end{bmatrix}, \begin{bmatrix} .78 & -.43 \\ 0.003 & 0 \end{bmatrix}, \begin{bmatrix} .18 & -.81 \\ .55 & 0 \end{bmatrix}$$

4. We now train on

$$\begin{bmatrix} 1 & 0 \\ 0 & 0 \end{bmatrix}, \begin{bmatrix} 0 & 1 \\ 0 & 0 \end{bmatrix}, \begin{bmatrix} 0 & 0 \\ 1 & 0 \end{bmatrix}, \begin{bmatrix} 0 & 0 \\ 0 & 1 \end{bmatrix}$$

After training, the resulting eigenvalues are $1, 1, 1, 1$. The 4 eigenvectors are

$$\begin{bmatrix} .64 & .22 \\ .04 & -.71 \end{bmatrix}, \begin{bmatrix} .67 & -.34 - 0.21i \\ .08 + .25i & .14 - .54i \end{bmatrix}, \begin{bmatrix} .67 & -.34 + 0.21i \\ .08 - .25i & .14 + .54i \end{bmatrix}, \begin{bmatrix} -.22 & .29 \\ .76 & .62 \end{bmatrix}$$

In each of the above examples, we have thus demonstrated that when training on $k$ linearly independent examples, $A_X$ obtains rank $k$.

Now we show that when $X$'s span has dimension $k$, then $A_X$ has rank $k$. To do this, we will consider a selection of $k$ basis elements from the above 4 matrices, and then show that introducing linear combinations of these basis elements into $X$ does not affect the rank of $A_X$.

Namely, we consider selecting the training set:

$$\begin{bmatrix} 1 & 0 \\ 0 & 0 \end{bmatrix}, \begin{bmatrix} 0 & 1 \\ 0 & 0 \end{bmatrix}, \begin{bmatrix} 0 & 0 \\ 1 & 0 \end{bmatrix}, \begin{bmatrix} 1 & 1 \\ 0 & 0 \end{bmatrix}$$

which has dimension 3. After training, the resulting eigenvalues are $1, 1, 1, -0.005$. The corresponding top 3 eigenvectors are:

$$\begin{bmatrix} .9 & -.28 \\ .33 & 0 \end{bmatrix}, \begin{bmatrix} .7 & .63 \\ -.33 & 0 \end{bmatrix}, \begin{bmatrix} -.03 & -.52 \\ -.85 & 0 \end{bmatrix},$$

Hence we again see that the rank of $A_X$ is given by the dimension of the span of the training set.

## D  LARGER TRAINING SETS FROM CIFAR10

In this section, we examine the operator of $D_{64,128}$ when trained using a larger dataset. Suppose we first train network $D_{64,128}$ for $20,000$ iterations using 10 examples, 1 from each class of CIFAR10 with training examples shown in Figure 9a. Network $D_{64,128}$ achieves an MSE of $0.0046$ after $20,000$ iterations.

In Figure 9c, there are 10 eigenvalues with magnitude close to 1. These top ten eigenvalues are the following:

1. $\lambda_1 = 1.04$
2. $\lambda_2 = 1.03$
3. $\lambda_3 = 1.00$
4. $\lambda_4 = 0.99$
5. $\lambda_5 = 0.98$
6. $\lambda_6 = 0.97 + 0.003i$
7. $\lambda_7 = 0.97 - 0.003i$

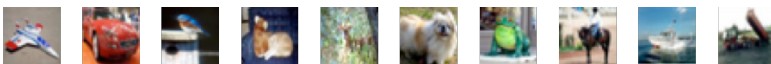

(a) 10 training examples from CIFAR10 with 1 example per class.

Top 64 Eigenvector Real Components     Top 64 Eigenvector Imaginary Components

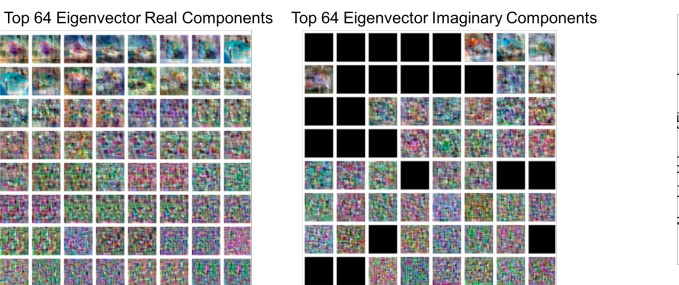

Plot of Eigenvalue Magnitudes

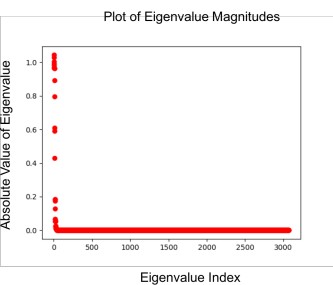

(b) The real and imaginary components of the eigenvectors corresponding to the top 64 eigenvalues of the operator of $D_{64,128}$ trained on the set in Subfigure 9a.

(c) The magnitude of eigenvalues of the operator of $D_{64,128}$.

Figure 9: Training set, eigenvector visualization, and plot of magnitude of eigenvalues for $D_{64,128}$ trained on 10 examples from CIFAR10.

8. $\lambda_8 = 0.96 + 0.045i$

9. $\lambda_9 = 0.96 - 0.045i$

10. $\lambda_{10} = 0.89$

Figure 9b presents the real and imaginary components of 64 eigenvectors corresponding to the top 64 eigenvalues (the grids should be read in row-major order). However, as the eigenvectors are now linear combinations of the training input, it is much harder to decipher exactly how the training examples are presented in the first 10 eigenvectors. However, we can clearly see that some of the training examples are present in these eigenvectors. For example, the real components of eigenvectors 1, 2, and 3 contain the bird training example, but with inverted colors. Similarly, the real components of eigenvectors 4 and 5 contain the the boat example again with inverted colors; the real components of eigenvectors 6 and 7 contain the frog example; the real components of eigenvectors 8 and 9 contain the car example.

Next we consider the rank of the learned operators when training on thousands of examples from a single class of CIFAR10. In particular, we trained using 2000 dogs, 5000 dogs, or 5000 planes from CIFAR10's training set. If our conjecture holds, then the resulting solution should be of rank equal to the dimension of the span of each of these training sets. Note that the rank of the learned solution should be less than 3072 even though we are training on greater than 3072 images because it is unlikely that there are 3072 linearly independent images in each of the sets. The training losses on each of these datasets is shown in Table 1 and the corresponding eigenvalue and eigenvector plots are shown in Figure 10.

From the plot of eigenvalues, it appears that the rank of the resulting matrix for $D_{64,192}$ on 2000 dogs, 5000 dogs, and 5000 planes is around $500, 900, 750$ respectively. However, it could be the case that the remaining eigenvalues are providing small nonzero contributions to the learned solution. To rectify this issue, we took the SVD of the resulting operator and zeroed out the lowest singular values and compared the MSE of the reconstructions using the resulting operator against those of the original. In particular, we compared the MSE of the reconstructions (after min-max scaling) to the training set used. The results are summarized in the Table 1.

From this table it is evident that only the top $500, 900$, and $750$ components were necessary to achieve low training error. Assuming our conjecture holds, this would indicate that these values are approximations for the dimension of the span of the corresponding training sets. We note that this is an approximation because the true dimension would be given when the training error approaches $0$. The training errors here are of the order of $10^{-3}$, thereby justifying such approximations.

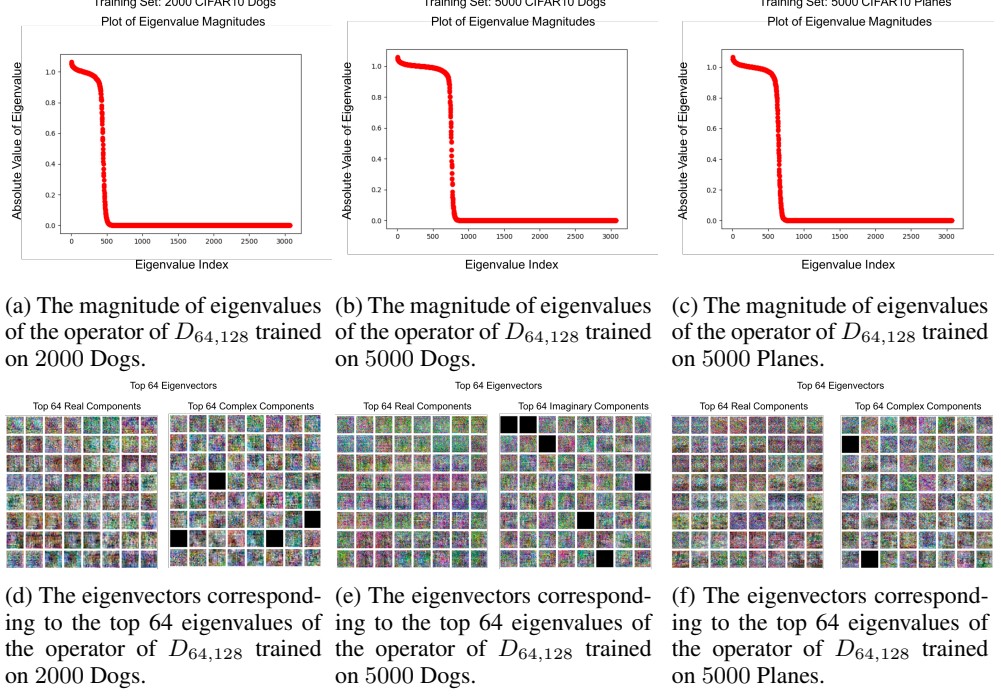

(a) The magnitude of eigenvalues of the operator of $D_{64,128}$ trained on 2000 Dogs.

(b) The magnitude of eigenvalues of the operator of $D_{64,128}$ trained on 5000 Dogs.

(c) The magnitude of eigenvalues of the operator of $D_{64,128}$ trained on 5000 Planes.

(d) The eigenvectors corresponding to the top 64 eigenvalues of the operator of $D_{64,128}$ trained on 2000 Dogs.

(e) The eigenvectors corresponding to the top 64 eigenvalues of the operator of $D_{64,128}$ trained on 5000 Dogs.

(f) The eigenvectors corresponding to the top 64 eigenvalues of the operator of $D_{64,128}$ trained on 5000 Planes.

Figure 10: Magnitude of eigenvalue plots and eigenvector visualizations for $D_{64,128}$ trained on 2000 Dogs, 5000 Dogs, and 5000 Planes.

# E    DIFFERENT INITIALIZATIONS IN NON-LINEAR SETTING

In Figure 11, we present the effect of different initializations on learning the point map in a nonlinear downsampling network ($NLD_{128,128}$). Note that $NLD_{128,128}$ learns the point map under any of Xavier/Kaiming uniform/normal initializations. However, interestingly, we note that the Kaiming initializations cause the network to learn a noisy version of the point map.

| Number of Components | Training Set | MSE on Training Set |
|---|---|---|
| All Components | 2000 CIFAR10 Dogs | 0.00347 |
| Top 500 Components | 2000 CIFAR10 Dogs | 0.00348 |
| All Components | 5000 CIFAR10 Dogs | 0.00219 |
| Top 900 Components | 5000 CIFAR10 Dogs | 0.00219 |
| All Components | 5000 CIFAR10 Planes | 0.00286 |
| Top 750 Components | 5000 CIFAR10 Planes | 0.00286 |

Table 1: A comparison of MSE (after min-max scaling) between using all singular values of the resulting operator of $D_{64,192}$ and only a subset of the singular values when trained on the listed training set.

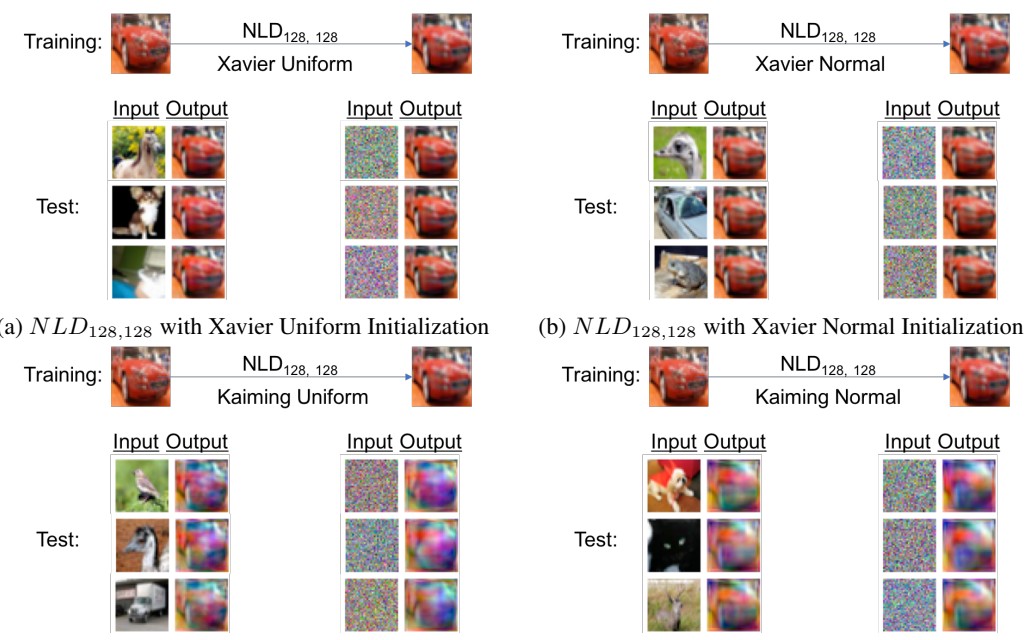

(a) $NLD_{128,128}$ with Xavier Uniform Initialization

(b) $NLD_{128,128}$ with Xavier Normal Initialization

(c) $NLD_{128,128}$ with Kaiming Uniform Initialization

(d) $NLD_{128,128}$ with Kaiming Normal Initialization

Figure 11: The nonlinear downsampling network ($NLD_{128,128}$) learns the point map regardless of the initialization used. However, we note that for Kaiming intializations, the resulting reconstruction after feeding in new inputs is a blurred version of the training image.

