# OpenReview forum: "Downsampling leads to Image Memorization in Convolutional Autoencoders"
_ICLR.cc/2019/Conference_

### Official Review · AnonReviewer1 · 2018-11-06
**Interesting idea but stronger supporting theory and more clarity are needed**

**Rating:** 5
**Confidence:** 2

**Review:**

The paper tries to provide an explanation for a memorization phenomenon observed in convolutional autoencoders. In the case of memorization, the autoencoder always outputs the same fixed image for any input image, even when the input image is random noise. The authors provide an empirical analysis that connects such a phenomenon to strides in convolutional layers of the autoencoder. Then, a possible theoretical explanation is given in the form of conjecture with some empirical evidence.

The paper presents very interesting idea, however presentation and theoretical foundation can be significantly improved.

- Please elaborate on how different initializations influence memorization effect. Currently the paper only mentions initialization approaches for which memorization can or cannot occur without going into deeper analysis.
- Having linear operator extraction described in the paper somehow breaks the flow, please consider moving it to Appendix.
- The comment after the Proposition section is not very clear. What does it mean that the Proposition does not imply that A_X must obtain rank which is given in the Conjecture? Please explain how is Proposition providing any theoretical support for Conjecture then.

- Minor comments
1. “2000 iteration” -> “2000 iterations”
2. The text says “Network ND trained on frog image” while the following next sentence says that “the network reconstructed the digit 3”. Please clarify.
3. “Network ND reconstructed the digit 3 with a training loss of 10^-4 and Network ND with loss 10^-2”. It seems that one of these should be “Network D”.
4. “(with downsamling)” ->  “(with downsampling)”

---

> ### Author Response · Authors · 2018-11-26
> **Response to Reviewer 1**
>
> We answer the reviewer questions in the points below:
>
> " - Please elaborate on how different initializations influence memorization effect. Currently the paper only mentions initialization approaches for which memorization can or cannot occur without going into deeper analysis."
>
> For the scope of our paper, we chose to only consider those initialization schemes that are most often used in practice i.e. Pytorch’s default uniform initialization, Xavier Uniform/Normal, and Kaiming Uniform/Normal.  In Figure 7 in the main text and in Appendix E, we display that for nonlinear downsampling networks, memorization occurs under any of these initialization schemes.
>
> " - Having linear operator extraction described in the paper somehow breaks the flow, please consider moving it to Appendix. "
>
> We thank the reviewer for the feedback on the flow of our paper.  The extraction of the linear operator is an important contribution to this work; so we felt it would be nice to have it in the main text, but we will make a smoother transition between the sections.
>
> "- The comment after the Proposition section is not very clear. What does it mean that the Proposition does not imply that A_X must obtain rank which is given in the Conjecture? Please explain how is Proposition providing any theoretical support for Conjecture then. "
>
> We wrote the comment here to emphasize that the Proposition mainly provides strong empirical support for the conjecture as opposed to a full proof of our conjecture on 2 x 2 images.  Namely, for any dataset of 2 x 2 images on which we trained the network in Figure 4a, we found that our conjecture held.
>
> We also thank the reviewer for pointing out the minor edits to our paper.

---

### Official Review · AnonReviewer3 · 2018-11-12
**Interesting direction but needs more to be a fully fleshed out paper.**

**Rating:** 5
**Confidence:** 2

**Review:**

Summary:-
The authors investigates downsampling as one method by which autoencoding CNNs may memorize data. The theoretical motivation provided concentrates on linear CNNs. They show that downsampling linear CNNs tent to learn a point-map of the training data, even though (under certain initializations) they are capable of learning identity maps. However, non-downsampling linear CNNs learn identity maps. Given enough data however, the authors claim that the downsampling CNN will learn the identity map.

Strengths:-
+ Authors present a good exploration of how linear CNNs memorize data when they do downsampling.
+ A theoretical prediction of the amount of training data needed to counteract data memorization for downsampling linear CNNs is provided, "Our conjecture also implies that when training a linear downsampling CNN on images of size 3 · 224 · 224, which corresponds to the input image size for VGG and ResNet (He et al. (2016), Simonyan & Zisserman (2015)), the number of linearly independent training examples needs to be at least 3 · 224 · 224 = 153, 228 before the network can learn the identity function."

Weaknesses:-
+ Not enough theoretical proof is provided to support the hypothesis. Which would be fine but some key experiments are missing to make the paper empirically rigorous.
++ Would be good to see experiments that illustrate the predication that a certain amount of data would allow for learning identity maps, both for linear and non-linear CNNs.
++ In the non-linear CNN setting, I'd like to see the same early-stopping experiment done for linear CNNs whose results are in Fig. 3. I don't see any obvious theoretical reason why that result form Fig. 3 must extend to the non-linear setting.
+ Initializations are pointed to as effecting the type of function the network learns. The authors give an example of a hand-designed initialization that allows a downsampling linear CNN to learn the identity map but they don't explain how they arrived at this initialization, or its properties that make it a good initialization. In general however, I think it's alright to assign exploration of effect of initialization to future work, since it seems like a non-trivial task.
+ It is mentioned that "the results are not observed for linear networks when using Kaiming initialization," which I read to mean the downsampling linear CNNs with Kaiming initialization learn the identity map, not point-map. If this is true, it seems like a vital point and should be included in discussions of future work.

Recommendation:  I think this could be a better short paper. There are some interesting contributions, but maybe not enough for a full length paper. For a full length paper, some further exploration of _why_ downsampling leads to (if indeed there is a causality) data memorization is needed.

Minor stuff:-
Citation "Gunasekar et al." is missing year (conclusions section)

---

> ### Author Response · Authors · 2018-11-26
> **Response to Reviewer 3**
>
> In the following we provide a point-by-point response to the remarks concerning weaknesses :
>
> In response to the first two points:  "+ Not enough theoretical proof is provided to support the hypothesis. Which would be fine but some key experiments are missing to make the paper empirically rigorous. " "++ Would be good to see experiments that illustrate the predication that a certain amount of data would allow for learning identity maps, both for linear and non-linear CNNs. "
>
> For linear downsampling networks, this experiment is shown in the proof of our proposition, namely we see that for 4 linearly independent training examples the network from Figure 4a learns the identity function exactly.  Depending on the nonlinearity it may be impossible for non-linear CNNs to learn the identity map (for example a single layer with a ReLU activation can never learn the full identity function as it will always zero out the negative values), which is why our conjecture is made only for linear downsampling networks.
>
> In response to "++ In the non-linear CNN setting, I'd like to see the same early-stopping experiment done for linear CNNs whose results are in Fig. 3. I don't see any obvious theoretical reason why that result form Fig. 3 must extend to the non-linear setting. "
>
> We have run these experiments and they are identical to those for linear networks in Figure 3.  For example, even after running for 100 iterations, downsampling non-linear networks learn the point map while non-downsampling non-linear networks learn a mapping more similar to the identity function.
>
> With regard to  "+ Initializations are pointed to as effecting the type of function the network learns. The authors give an example of a hand-designed initialization that allows a downsampling linear CNN to learn the identity map but they don't explain how they arrived at this initialization, or its properties that make it a good initialization. In general however, I think it's alright to assign exploration of effect of initialization to future work, since it seems like a non-trivial task."
>
> In regard to the initialization of the identity map, we provide a full description of the initialization in Appendix A.  To provide more intuition around how we arrived at this initialization: the manual initialization is meant to preserve all pixels of the input image through the layers so that they can be rearranged by the final layer to get the identity function.
>
> Lastly, in response to "+ It is mentioned that "the results are not observed for linear networks when using Kaiming initialization," which I read to mean the downsampling linear CNNs with Kaiming initialization learn the identity map, not point-map. If this is true, it seems like a vital point and should be included in discussions of future work."
>
> We thank the reviewer for pointing out this possible misinterpretation. When we say that the results are not observed for linear networks when using the Kaiming initialization, we meant to say that under the Kaiming initialization, the downsampling linear CNN learns neither the point map nor the identity map.
>
>
> "Recommendation: I think this could be a better short paper. There are some interesting contributions, but maybe not enough for a full length paper. For a full length paper, some further exploration of _why_ downsampling leads to (if indeed there is a causality) data memorization is needed."
>
> Thanks, we will consider this in our future submission.
>
> "Minor stuff:- Citation "Gunasekar et al." is missing year (conclusions section)"
>
> Agreed, thanks.

---

### Official Review · AnonReviewer2 · 2018-11-13
**Promising idea that requires substantial improvement in analysis and presentation**

**Rating:** 3
**Confidence:** 3

**Review:**

The authors conjecture that convolutional downsampling is an underlying mechanism behind sample memorization in over-parameterized convolutional autoencoders. They claim that this effect leads the system to converge to a low-rank solution in contrast to the theoretically possible identity mapping. They support their claim with numerical experiments on linear and non-linear convolution autoencoders.

Strengths:
- The authors develop their idea in close connection to commonly used architectures.

Weaknesses:
- The main statements concern the architecture; however, the experiments do not account for the many confounding factors such as initialization or the chosen optimizer. The paper itself states on page 4 that the results depend on the initialization and cite Gunasekar et al. in the conclusion for an analogy, which, however, explores the implicit regularization effect of a gradient descent optimizer.
- There is no clear and proved statement despite the suggestive mathematical nature of the writing (Conjecture, Proposition). The claimed 'proof' of the Proposition is conducted via experiment. In light of the above mentioned confounding factors, the current phrasing of the Proposition will not allow a formal proof as it is unclear what the system 'linear Network DS' even is.
- The boundary between conjectured and inferred statements is very vague. For example, the meaning of  'prefers to learn a point map' is unclear.

Overall, the exposition is insufficient in supporting the conjectured effect. The methodology could be strengthened in two directions:
1) experimentally: designing numerical experiments that exclude confounding factors and surface the conjectured effect
2) theoretically: abstracting the idea into a clear mathematical statement that can be proved

I encourage the authors to extend their work for submission to a future venue.

---

> ### Author Response · Authors · 2018-11-26
> **Response to Reviewer 2 Comments**
>
> Addressing the first remark:  "- The main statements concern the architecture; however, the experiments do not account for the many confounding factors such as initialization or the chosen optimizer. The paper itself states on page 4 that the results depend on the initialization and cite Gunasekar et al. in the conclusion for an analogy, which, however, explores the implicit regularization effect of a gradient descent optimizer."
>
> In regard to initialization: we state on page 4 and in Appendix E that nonlinear downsampling networks learn the point-map (i.e. a low rank solution) under any of the popular initializations (Pytorch’s default uniform, Xavier normal and uniform, and Kaiming normal and uniform), which should eliminate initialization as a confounding factor.  In the conclusion, our citation of Gunasekar et al. is to emphasize that they observe that gradient descent learns a low rank solution for the problem of matrix factorization just as we have observed that gradient descent learns a low rank solution (the point map) for downsampling autoencoders.   We do not draw any analogies to our remarks on initialization in regard to this paper.
>
> Addressing the second remark: " - There is no clear and proved statement despite the suggestive mathematical nature of the writing (Conjecture, Proposition). The claimed 'proof' of the Proposition is conducted via experiment. In light of the above mentioned confounding factors, the current phrasing of the Proposition will not allow a formal proof as it is unclear what the system 'linear Network DS' even is."
>
> The proposition claims that our linear downsampling network (under the default Pytorch initialization) defined in Figure 4a can learn a solution of rank 1, 2, 3 or 4, which we demonstrate empirically.  As we state in the paper, this is not a proof for our conjecture for images of size 2 x 2, but rather it provides strong empirical evidence that our conjecture holds for this network acting on 2 x 2 images.
>
> Addressing the last remark: "- The boundary between conjectured and inferred statements is very vague. For example, the meaning of 'prefers to learn a point map' is unclear. "
>
> We respectfully disagree in regard to the vagueness between our conjectured and inferred statements.  The example provided by the reviewer here is for a sentence in our conclusion where we state that “downsampling CNN architecture[s] with the capacity to learn the identity function prefer[s] to learn a point map.”  As demonstrated in Figure 4, we are merely emphasizing the point that downsampling CNNs learn the point map even though they have the capacity to learn the identity function.

---

### Author Response · Authors · 2018-11-26
**Reviewer Overall Response**

We thank the three reviewers for their comments.  We have provided a point-by-point response under each of the reviews.

---

### Meta-Review · Area_Chair1 · 2018-12-17
**Insufficient Rigor**

**Confidence:** 5
**Recommendation:** Reject

**Metareview:**

This paper studies the question of memorization within overparametrised neural networks. Specifically, the authors conjecture that memorization is linked to the downsampling operators present in many convolutional autoencoders.

All reviewers agreed that this is an interesting question that deserves further analysis. However, they also agreed that in its current form, the paper lacks mathematical and experimental rigor. In particular, the paper does not follow the basic mathematical standards of proving any stated proposition/theorem, instead mixing empirical with mathematical proofs. The AC fully agrees with the points raised by reviewers, and therefore recommends rejection at this point, encouraging the authors to address these important points before resubmitting their work.